# Association between inflammatory biomarkers and cognitive aging

Yuan Fang[1]*, Margaret F. Doyle[2], Jiachen Chen[1], Michael L. Alosco[3,4], Jesse Mez[3,4,5], Claudia L. Satizabal[4,6], Wei Qiao Qiu[3,7,8], Joanne M. Murabito[5,9,10], Kathryn L. Lunetta[1]

1 Department of Biostatistics, School of Public Health, Boston University, Boston, Massachusetts, United States of America, 2 Department of Pathology and Laboratory Medicine, Larner College of Medicine, University of Vermont, Burlington, Vermont, United States of America, 3 Boston University Alzheimer's Disease Research Center and CTE Center, School of Medicine, Boston University, Boston, Massachusetts, United States of America, 4 Department of Neurology, School of Medicine, Boston University, Boston, Massachusetts, United States of America, 5 Framingham Heart Study, National Heart, Lung, and Blood Institute and Boston University School of Medicine, Framingham, Massachusetts, United States of America, 6 Glenn Biggs Institute for Alzheimer's and Neurodegenerative Diseases, University of Texas Health Science Center at San Antonio, San Antonio, Texas, United States of America, 7 Department of Psychiatry, School of Medicine, Boston University, Boston, Massachusetts, United States of America, 8 Department of Pharmacology & Experimental Therapeutics, School of Medicine, Boston University, Boston, Massachusetts, United States of America, 9 Department of Medicine, Section of General Internal Medicine, School of Medicine, Boston University, Boston, Massachusetts, United States of America, 10 Boston Medical Center, Boston University, Boston, Massachusetts, United States of America

* yuanf@bu.edu

**Data Availability Statement:** The data that support the findings of this study are available from the Framingham Heart Study through NHLBI's Biologic Specimen and Data Repository Information

## Abstract

Inflammatory cytokines and chemokines related to the innate and adaptive immune system have been linked to neuroinflammation in Alzheimer's Disease, dementia, and cognitive disorders. We examined the association of 11 plasma proteins (CD14, CD163, CD5L, CD56, CD40L, CXCL16, SDF1, DPP4, SGP130, sRAGE, and MPO) related to immune and inflammatory responses with measures of cognitive function, brain MRI and dementia risk. We identified Framingham Heart Study Offspring participants who underwent neuropsychological testing (n = 2358) or brain MRI (n = 2100) within five years of the seventh examination where a blood sample for quantifying the protein biomarkers was obtained; and who were followed for 10 years for incident all-cause dementia (n = 1616). We investigated the association of inflammatory biomarkers with neuropsychological test performance and brain MRI volumes using linear mixed effect models accounting for family relationships. We further used Cox proportional hazards models to examine the association with incident dementia. False discovery rate p-values were used to account for multiple testing. Participants included in the neuropsychological test and MRI samples were on average 61 years old and 54% female. Participants from the incident dementia sample (average 68 years old at baseline) included 124 participants with incident dementia. In addition to CD14, which has an established association, we found significant associations between higher levels of CD40L and myeloperoxidase (MPO) with executive dysfunction. Higher CD5L levels were significantly associated with smaller total brain volumes (TCBV), whereas higher levels of sRAGE were associated with larger TCBV. Associations persisted after adjustment for *APOE* ε4 carrier status and additional cardiovascular risk factors. None of the studied inflammatory biomarkers were significantly associated with risk of incident all-cause dementia. Higher

Coordinating Center (BioLINCC). https://biolincc.nhlbi.nih.gov/studies/fhs/.

**Funding:** This work was supported by the National Heart, Lung, and Blood Institute and the Boston University School of Medicine, Framingham Heart Study (contract number 75N92019D00031); and by the National Institute on Aging (grant number R01AG067457 (awarded to MFD, JMM, and KLL), U19AG068753, P30AG072978 (awarded to MLA)); and National Institutes of Neurological Disorders and Stroke (grant number K23NS102399 (awarded to MLA)). The Neuropsychological testing data and brain MRI data were collected with funding from the National Institutes on Aging and Neurological Disorders and Stroke grants (grant number R01AG016495, R01AG008122, R01AG033040, R01AG054076, R01AG049607, R01AG033193, and R01NS017950).

**Competing interests:** The authors have declared that no competing interests exist.

circulating levels of soluble CD40L and MPO, markers of immune cell activation, were associated with poorer performance on neuropsychological tests, while higher CD5L, a key regulator of inflammation, was associated with smaller total brain volumes. Higher circulating soluble RAGE, a decoy receptor for the proinflammatory RAGE/AGE pathway, was associated with larger total brain volume. If confirmed in other studies, this data indicates the involvement of an activated immune system in abnormal brain aging.

## Background

Systemic inflammation plays a key role in the disease pathology of Alzheimer's Disease (AD) and related dementias [1]. Existing literature has examined various inflammatory and cellular immunity biomarkers related to cognitive and brain aging. Chronic peripheral inflammation, as measured by C-reactive protein, is associated with an increased risk of dementia, including AD dementia, and MRI-derived biomarkers of brain atrophy among persons with increased genetic susceptibility to AD [2]. A composite score of five neutrophil-related inflammatory biomarkers (neutrophil gelatinase-associated lipocalin (NGAL), myeloperoxidase (MPO), interleukin-8 (IL-8), macrophage inflammatory protein-1 beta (MIP-1β), and tumor necrosis factor (TNF)), predicted accelerated decline in executive function over one year among people with AD dementia [3]. Markers of monocyte activation, such as monocyte differentiation antigen CD14 (sCD14), associate with increased risk for incident dementia, and with cognitive function and brain MRI markers of brain aging in two community based cohorts [4], while soluble CD163 is elevated in inflammatory diseases [5] and subarachnoid hemorrhage [6]. Activation of immune cells can release enzymes that can generate soluble forms of receptors, such as sCD40L, sCD163, sCD14, sCD56 (NCAM), sGP130, sRAGE, sCXCL16, and DPP4, all of which have been implicated in dementia and AD [7–15]. Additionally, soluble factors that are upregulated upon activation of the immune system, such as stromal-derived factor 1 (SDF-1, CXCL12) and CD5L, play a role in immune cell recruitment to sites of lesions and facilitate removal of damage associated molecular patterns (DAMP), both of which play a role in the pathobiology of AD and related dementias. Inflammatory markers are also associated with greater brain atrophy than expected for age alone [16]. Advanced glycation end products are known to be associated with inflammatory responses and may be associated with diseases of aging including dementia. In the Rotterdam Study, low soluble receptor for advanced glycation end products (sRAGE) was associated with higher prevalence of dementia but not with dementia incidence [17]. In the Mayo Clinic Study of Aging, inflammatory cytokine levels in plasma were not associated with cross-sectional or longitudinal global or domain-specific cognitive test scores, but did associate with diagnosis of mild cognitive impairment [18]. Meta-analyses of human observational studies have supported the role of some inflammatory biomarkers in increased risk of all-cause dementia, with smaller or non-significant effects for AD dementia [19].

Further investigation of circulating inflammatory biomarkers with cognitive function and brain imaging endophenotypes and dementia risk in large community-based samples is needed to determine if inflammatory biomarkers may be effective targets for prevention or intervention of dementia and AD dementia.

We propose to investigate the association of 11 inflammatory protein biomarkers with cognitive function, incident dementia, and brain MRI measures of brain atrophy among Framingham Heart Study (FHS) Offspring cohort participants. We hypothesize that individual

biomarkers will associate cross-sectionally with cognitive function and brain imaging measures, and longitudinally with risk of incident dementia. Further, because the *APOE* protein is involved in inflammation [20] and immunoregulation [21], among other biological processes, we tested whether *APOE* carrier status is an effect modifier of these associations.

## Methods

### Study sample

The FHS is a community-based prospective cohort study that recruited 5209 participants as the Original cohort in 1948 [22]. The Offspring cohort, recruited in 1971 (n = 5129), includes adult children who had at least one parent in the Original cohort and their spouses [23, 24]. Offspring participants have received examinations once every 4–8 years since enrollment. There were 3539 Offspring participants who attended the seventh examination (1998–2001), 3295 of whom provided a blood sample for quantifying the protein biomarkers of interest. Details of the sample selection for dementia, neuropsychological testing and brain MRI are provided below in subsections describing each of the outcome sets and in a flow chart in Fig 1.

All participants provided written informed consent at the time of each attended FHS examination; existing data was used for this study. FHS exams were reviewed and approved by the Institutional Review Board (IRB) at Boston University Medical Center (BUMC). The BUMC

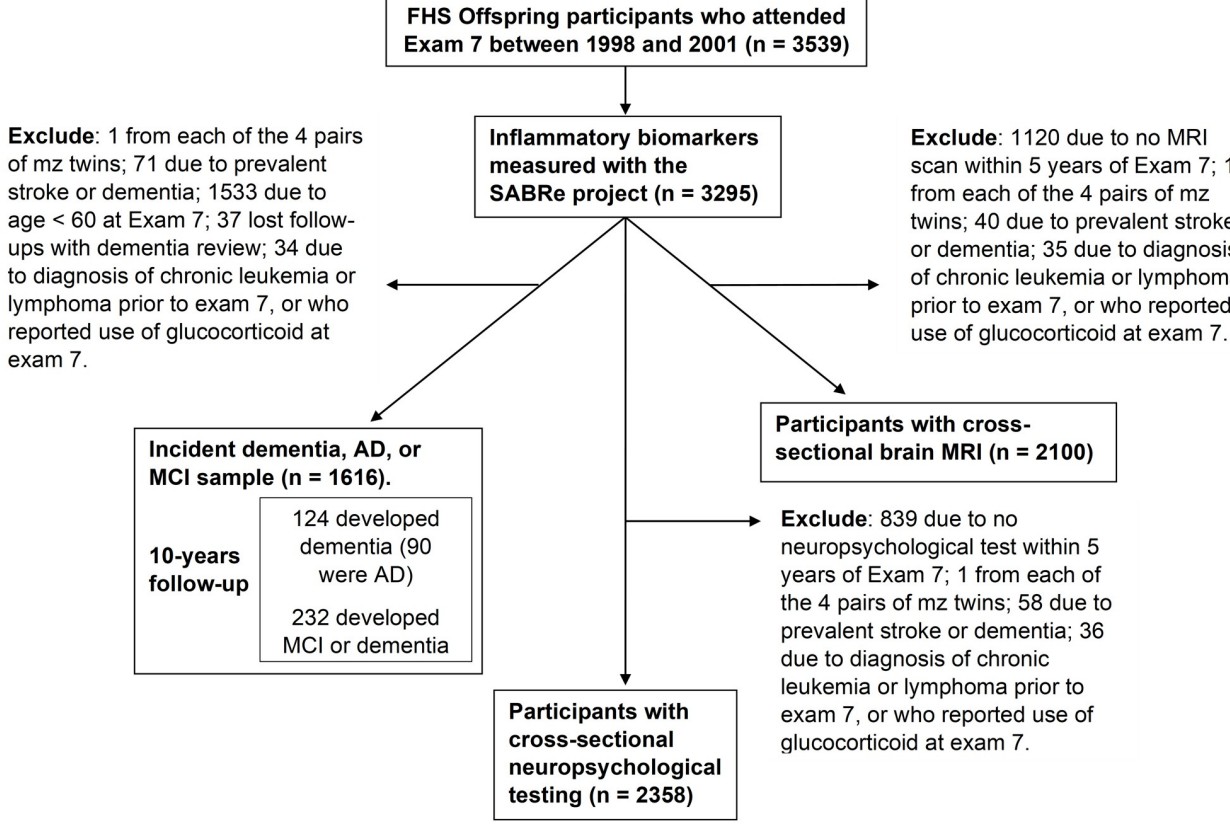

**Fig 1. Study sample for the incident dementia, neuropsychological testing and the brain MRI outcomes.** The biomarkers were collected from the Offspring cohort examination 7(1998–2001). One from each of 4 pairs of Monozygotic (mz) twins with biomarkers measurements were excluded. Participants with prevalent dementia or stroke, participants diagnosed with chronic leukemia or lymphoma prior to exam 7, and participants who reported use of glucocorticoid medication at exam 7 were excluded from all study samples. A 10-year follow-up window was applied in the study for incident dementia, AD or MCI. The closest neuropsychological testing battery and brain MRI scan within five years of the examination was used.

IRB number for this project is H-39876; and the current BUMC IRB number for FHS is H-32132.

## Inflammatory protein biomarkers

The Systems Approach to Biomarker Research in Cardiovascular Disease initiative measured 85 plasma proteins in the participants of the FHS Offspring cohorts [25]. From these, we chose 11 protein biomarkers related to innate or adaptive immune cells or known to be associated with inflammatory responses for our investigation, as mounting evidence demonstrates a role for immune cells and inflammation in the disease pathogenesis of AD dementia and cognitive disorders [26]. The biomarkers included are: sCD14, scavenger receptor cysteine-rich type 1 protein (CD163), CD5 molecule-like (CD5L), neural cell adhesion molecule (CD56), soluble CD40 ligand (CD40L), chemokine (C-X-C motif) ligand 16 (CXCL16), stromal cell-derived factor 1 (SDF1), dipeptidyl-peptidase 4 (DPP4), interleukin-6 receptor beta (SGP130), sRAGE, and myeloperoxidase (MPO). CD14 serves as a positive control as previous work demonstrates an association between sCD14 and dementia in FHS [4]. Details on protein biomarker measurement have been described in previous publications [27, 28]. In brief, the proteins were measured by the Systems Approach to Biomarker Research (SABRe) that was established by the National Heart, Lung, and Blood Institute [27]. Target proteins were measured from frozen fasting plasma samples and were assayed using a modified ELISA sandwich method, multiplexed on a Luminex xMAP platform (Sigma-Aldrich, St. Louis, MO). Methods for antibody conjugation and multiplex assay development followed protocols recommended and developed by Luminex and were conducted by a contracted laboratory (Sigma-Aldrich). Quality control of each marker used both a "High" and "Low" spike control; inter-assay and intra-assay coefficients of variation (CV) were also quantified. Upper CV limits were applied to each assay to remove extreme CV outliers. The detectable ranges, as well as the upper CV limit of the 11 protein biomarkers selected for our analysis are summarized in the Supplementary document S1a Table). For each protein biomarker, values outside the detectable limits were set to the lower or upper detectable limit respectively and were set to unknown if the CV was above the threshold [27]. S1b Table) summarizes the number of cases when the protein biomarkers of interest are outside the detectable ranges, within the detectable range, above the upper CV threshold, or are missing. Hence the exact sample size for each protein biomarker is distinct. In our analyses, protein biomarkers were rank normalized to mean 0 and standard deviation (SD) 1.

## Assessment of cognitive function

Offspring participants who attended at least one core FHS examination between exam 5 (1991–1995) and exam 7 were invited to complete a battery of neuropsychological tests that were administered by a trained psychometrician using standard administration protocols [29]. For cognitive function outcomes in our cross-sectional analyses, we identified 2358 participants who completed a battery of neuropsychological tests within five years of exam 7 and did not have prevalent stroke or dementia at the earlier date between the exam 7 date or neuropsychological test date. We excluded participants with diagnosis of chronic leukemia or lymphoma prior to exam 7 or who reported use of glucocorticoid medication at exam 7. Details for sample selection are provided in Fig 1.

Participants were assessed using seven tests that assess function across four cognitive domains: the Wechsler Memory Scale [30] Logical Memory-II, Paired Association learning, and Visual Reproduction, delayed recall components (LMD, PASD, and VRD) assessing verbal and visual episodic memory; the Trails Making Tests Parts A and B [31, 32] and the Wechsler

Adult Intelligence Scale Fourth Edition [33] Similarities (SIM) subtest assessing attention and executive function; the Hooper Visual Organization Test (HVOT) [34] assessing visuoperceptual skills; and the 30-item Boston Naming Test (BNT30) [35] assessing language, specifically, confrontation naming. For Trail Making Tests, we used the difference between the time in minutes taken to complete parts B and A (denoted as TRAILSBA) [36]. Smaller values in the TRAILSBA score indicates better performance, whereas lower scores in all other tests reflect poorer performance. A few participants participated in neuropsychological testing prior to the one included in the study, which can lead to practice effects [37]. Therefore, an indicator "retest" of whether the neuropsychological testing battery was the first received by the participant was included for neuropsychological tests analyses.

## MRI assessment

Brain MRI techniques used in the FHS have been described previously [38–40]. In brief, brain MRI images were obtained on a 1 or 1.5 Tesla Magnetom Siemens scanner using 3D T1-weighted coronal spoiled gradient-recalled echo acquisition and T2-weighted double spin-echo (DSE) coronal sequences acquisition. All images were centrally read, blind to participants' demographic and clinical characteristics. The segmentation and the protocols for quantifying total and regional brain volumes, as well as white matter hyperintensity (WMH) volumes, have been described elsewhere [39–42]. The segmentation of brain volumes is based on an Expectation-Maximization algorithm [43]; hippocampal volume (HPV) is computed using a semi-automated multiatlas hippocampal segmentation algorithm [44]; segmentation of WMH from brain matter is based on a repeat Gaussian distribution to the summed image data (from the DSE sequences after removal of cerebral spinal fluid and correction of image intensity non-uniformities) and using a priori determined segmentation threshold [39].

We identified 2100 Offspring participants who participated in the brain MRI scan within five years of exam 7, excluding those with prevalent stroke or dementia, prevalent chronic leukemia or lymphoma prior to exam 7, and those reported to be on glucocorticoid medication at exam 7. Our primary outcomes included total cerebral brain volume (TCBV), HPV, and volume of WMH. We further hypothesized that the role of inflammation may have different regional gray matter associations [45–51]. Therefore, our secondary outcomes included five gray matter volumes: cerebral gray matter volume (CGV), frontal gray matter volume (FGV), temporal gray matter volume (TGV), parietal gray matter volume (PGV), and occipital gray matter volume (OGV). Total and regional brain volumes, as well as WMH, were computed as percentage of total intracranial volume (TCV) to correct for differences in head size [39, 43].

## FHS dementia ascertainment

Surveillance methods and dementia tracking for the FHS have been detailed elsewhere [52–55]. Briefly, general cognitive status for the Offspring cohort has been monitored and assessed using the Mini-Mental State Examination (MMSE) at each examination cycle beginning with Offspring exam 5 (1991–1995). Participants were flagged for dementia review if the MMSE performance fell below education-based cutoff scores at any examination, declined 3 or more points between consecutive examinations, or decreased 5 or more points from the participants highest past MMSE score. In addition, participants were also flagged for suspected cognitive impairment following referrals from FHS investigators or concern from the participants, their families, their doctors, or other outside practitioners. Once selected for dementia review, a participant's cognitive status was evaluated by a dementia review panel which included a neurologist and a neuropsychologist and was assigned a cognitive status of normal, mild cognitive impairment (MCI), or dementia. The panel also determined the dementia subtype and date of

diagnosis using data from multiple sources [53]. The Diagnostic and Statistical Manual of Mental Disorders, 4th Edition, Text Revision (DSM-IV-TR) criteria [56] and the National Institute of Neurological and Communicable Disease and Stroke-Alzheimer's Disease and Related Disorders Association (NINCDS-ADRDA) [57] were used for diagnoses of dementia and AD dementia respectively. Diagnosis of AD by NINCDS-ADRDA criteria includes AD with or without stroke, and mixed type of AD and vascular dementia. The Key Symposium Working Group on Mild Cognitive Impairment criteria [58] were used to for MCI.

Participants with age < 60 at exam 7 were excluded from our analysis for incident dementia since dementia is rare in younger participants [4, 55]. We also excluded participants with prevalent dementia at exam 7, those who were diagnosed with chronic leukemia or lymphoma prior to exam 7, and who reported to be on glucocorticoid medication at exam 7, resulting in a sample size of $n$ = 1616 for the incident dementia analysis.

*APOE* genotypes were determined by utilizing the polymerase chain reaction and restriction isotyping [59].

## Statistical analyses

We had three outcome sets: the neuropsychological test performance, the brain MRI measures, and incident dementia; we tested for pairwise associations between outcome measures from each of the outcome sets and each of the 11 inflammatory biomarkers. Neuropsychological test scores were rank normalized to mean 0 and SD 1. WMH volume was log transformed to normalize the skewness in its distribution. Two statistical models were considered in our analyses. Our primary model (Model 1) covariates included age, sex, and, for cognitive function and MRI, time between blood sample (exam 7) and measurement of cognitive function/MRI. For MRI outcomes, Model 1 also included age$^2$ and an age-sex interaction terms. For cognitive function and dementia outcomes, Model 1 additionally included educational level. For cognitive function, a retest indicator for the cognitive outcomes was also included. Education level for cognitive outcomes was recorded at the cognitive test; for the incident dementia analysis, education level was derived using the highest education level from the records at the Offspring second examination, the eighth examination, and the neuropsychological test. For both cases, education level was defined as a four-category variable (did not graduate high school; high school graduate; some college; college graduate).

Model 2 included all Model 1 covariates and additional adjustment for Apolipoprotein E (*APOE*) $\varepsilon$4 carrier status, cardiovascular disease (CVD) risk factors, prevalent atrial fibrillation (AF), and prevalent CVD. Specifically, the CVD risk factors included systolic blood pressure (SBP, mmHg), treatment for hypertension, body-mass index (BMI, kg/m$^2$), current smoking status, total cholesterol level (mg/dL), high density lipoprotein cholesterol level (HDL, mg/dL), and presence of diabetes. Presence of diabetes was determined if any of the following were satisfied: fasting blood glucose level of 126 mg/dL or higher, random blood glucose level of 198 mg/dL or higher or use of antidiabetic medications. Prevalent CVD was defined at the exam of the blood draw based on current or previous diagnosis of coronary heart disease (myocardial infarction, angina pectoris, coronary insufficiency), transient ischemic attack, intermittent claudication, and congestive heart failure determined by adjudication of a panel of senior investigators. All covariates except *APOE* $\varepsilon$4 carrier status and education level were directly measured at the same FHS exam as the blood draw.

In our primary analyses, cross-sectional association between the 11 biomarkers and cognitive function and brain MRI measures were investigated using linear mixed effect models adjusting for Model 1 covariates. Familial correlation among relatives was accounted for using a random effect with the kinship matrix. In addition, association between the 11 protein

biomarkers and incident all-cause dementia was investigated using Cox proportional hazards models, adjusting for Model 1 covariates. We considered a 10-year follow-up from the Offspring seventh examination (when biomarkers were measured). For participants with incident all-cause dementia, follow-up time was measured as years from the baseline examination (Exam 7) to the diagnosis. Among participants who did not develop dementia, we censored follow up at the last date they were know not to have dementia, date of death, date of last exam attended if within the follow-up window, or a maximum of 10 years from the baseline examination. Robust standard errors were used to account for correlation among related individuals. We report the hazard ratios (HRs) accompanied by 95% confidence intervals (CIs). Robustness of association between the biomarkers and cognitive function, brain MRI measures, and incident all-cause dementia to the additional adjustment of *APOE ε*4 status, CVD risk factors, and prevalent CVD and prevalent AF were examined using Model 2 covariates.

Our secondary analyses included the investigation of association of the protein biomarkers with neuropsychological testing performance, brain MRI measures, and incident dementia stratified by the *APOE ε*4 carrier status. Interaction between biomarker and *APOE ε*4 carrier status was first tested; then analyses were carried out separately on the carrier and non-carrier subgroups adjusting for Model 1 covariates.

A false discovery rate (FDR) [60] within each set of outcomes was computed to account for multiple testing, where the number of tests included in the FDR was 11 (protein biomarkers) multiplied by the outcomes included in the outcome set. A threshold of FDR $\leq$ 0.1 for each outcome set was used for declaring a significant association. All analyses were conducted in *R-4.0.2* software [61] using the *coxph* function [62] for the cox proportional hazard models and *lmekin* function [63] for the linear mixed effect models.

## Sensitivity analyses

We carried out the following sets of sensitivity analyses: 1) We examined the cross-sectional association between the biomarkers and the cognitive testing outcome and the brain MRI outcome adjusting for Model 1 covariates on subsamples of participants who had neuropsychological test or brain MRI scan within 2 years before or after the blood draw (N = 2185 and 1945, respectively). 2) We examined the association between the biomarkers and the risk of AD, a subtype of dementia. For participants who developed AD, follow-up time was measured as years from the baseline examination to the diagnosis. Among participants who were not diagnosed with AD, data were censored at the date of other types of dementia onset, date of death, date of last exam attended if within the follow-up window, or the end of the follow-up. 3) We added MCI to our primary time-to-event analysis outcome because MCI can be an early stage of the disease continuum of dementia. Specifically, follow-up time was measured in years from the baseline examination to the diagnosis of MCI or dementia; or censored at the time that participants were know not to be cognitive impaired, date of death, or date of last examination attended before the follow-up window ended.

## Results

### Participants characteristics

The overlap of the study sample for the three outcome sets is shown in S1 Fig. The three study subsamples are highly overlapped with each other: there were 2094 participants included in both the neuropsychological test subsample and the MRI subsample; among whom, 1038 were included in all three subsamples. Table 1 shows the participant demographics for the neuropsychological test, MRI, and dementia study subsamples. The neuropsychological test and MRI subsamples have an average age of 61 years. Due to the exclusion of participants with

**Table 1.  Participant characteristics of the incident dementia, neuropsychological (NP) test, and brain MRI subsamples.**

| | NP Sample | MRI Sample | Dementia Sample |
|---|---|---|---|
| | N = 2358 | N = 2100 | N = 1616 |
| **Incident dementia, n (%)** | - - - | - - - | 124 (8%) |
| **Time to dementia, years, mean (SD)** | - - - | - - - | 6 (3) |
| **Incident AD, n (%)** | - - - | - - - | 90 (6%) |
| **Incident MCI or dementia, n (%)** | - - - | - - - | 232 (14%) |
| **Age at Exam 7, years, mean (SD)** | 61 (9) | 61 (9) | 68 (6) |
| Female, n (%) | 1262 (54%) | 1123 (53%) | 861 (53%) |
| College graduated, n (%) | 914 (39%) | - - - | 518 (32%) |
| **Distance between NP/MRI and Exam 7, years, mean (SD)** | 0.8 (0.8) | 0.8 (0.8) | - - - |
| **Total Cholesterol, mg/dL, mean (SD)** | 201 (37) | 201 (37) | 199 (37) |
| **HDL, mg/dL, mean (SD)** | 54 (17) | 54 (17) | 53 (17) |
| **SBP mm Hg, mean (SD)** | 126 (18) | 126 (18) | 132 (19) |
| **BMI, kg/m², mean (SD)** | 28.0 (5.3) | 27.9 (5.2) | 28.2 (5.0) |
| **On treatment for hypertension, n (%)** | 724 (31%) | 628 (30%) | 713 (44%) |
| **Diabetes, n (%)** | 238 (10%) | 211 (10%) | 239 (15%) |
| **Current smoking, n (%)** | 294 (12%) | 262 (12%) | 144 (9%) |
| **Prevalent CVD, n (%)** | 246 (10%) | 207 (10%) | 268 (17%) |
| **Prevalent AF, n (%)** | 81 (3%) | 63 (3%) | 99 (6%) |
| *APOE ε4* **carriers, n (%)** | 522 (22%) | 472 (22%) | 360 (22%) |
| **First NP test included, n (%)** | 2339 (99%) | - - - | - - - |
| **MMSE at Exam 7, median (IQR)** | 29 (2) | 29 (2) | 29 (2) |

age < 60, the dementia outcome subsample has an average age of 68 years at exam 7. Each subsample included ~ 54% female participants; and the percentage of *APOE ε4* carriers in all subsamples was 22%. A total of 39% of the 2358 neuropsychological test subsample participants and 32% of the 1616 dementia subsample participants had a college degree. We observed a higher proportion of prevalent diabetes, prevalent CVD and AF for the dementia subsample participants compared to the two cross-sectional study subsamples, where the latter two were similar. Average total score on the MMSE was 29 (SD = 2) at baseline for the incident dementia study subsample and for the neuropsychological test and MRI subsamples. There were 124 cases of dementia within 10 years of follow-up, with an average time since Exam 7 to dementia of 6 years (SD = 3). Of the dementia cases, 90 were clinically consistent with AD. An additional 108 developed MCI, for a total of 232 (14%) who developed MCI or dementia within 10 years. The neuropsychological testing battery and brain MRI data were measured on average 0.8 years (SD = 0.8 years) after exam 7.

The mean and SD for the 11 protein biomarkers, as well as distributions of the neuropsychological testing scores and the brain MRI measurements are summarized in the supplementary materials S2 Table. Correlations among the cognitive scores and MRI measurements are provided in S2 Fig.

## Biomarkers and cognitive testing performance

We observed significant negative associations for CD14, CD40l, and MPO with performance on the SIM test of attention and executive function adjusting for model 1 covariates (Fig 2.), indicating that higher levels of these proteins is associated with poorer scores on this test. Effect estimates for all three proteins were similar, ($\beta$ = −0.060, −0.070 − 0.061, S3 Table). We also observed positive association for sRAGE with the SIM test within effect estimate of

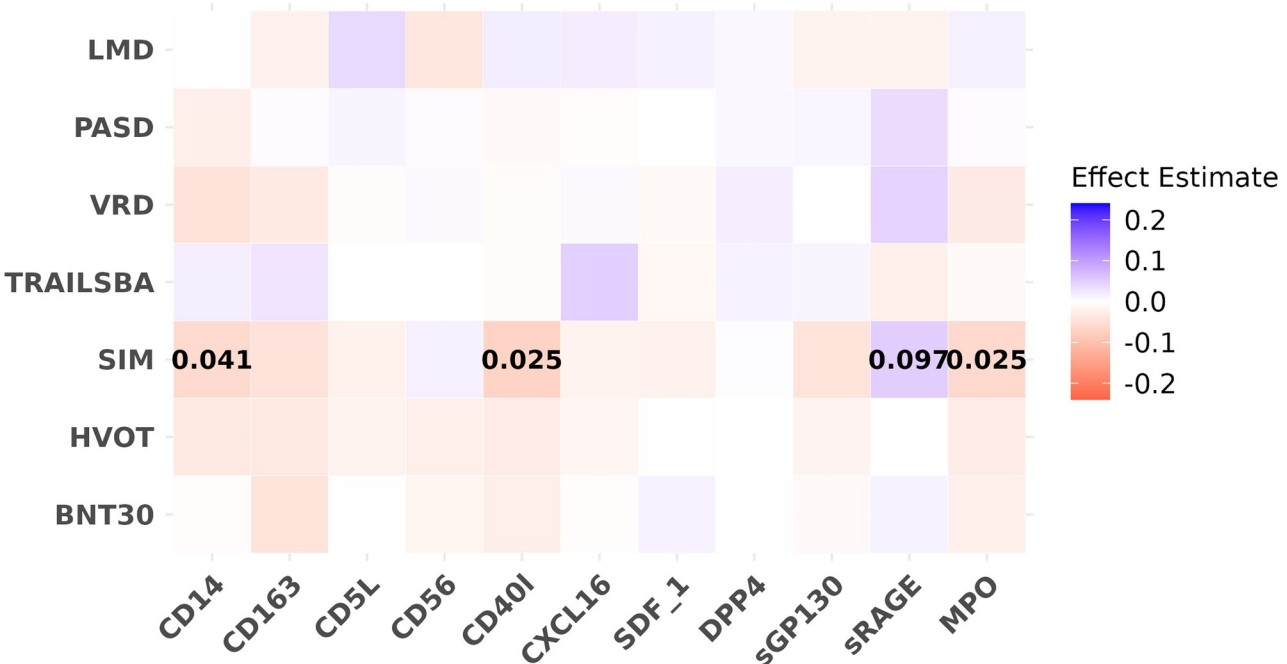

**Fig 2. Effect estimates and FDR value for the associations of protein biomarkers with neuropsychological testing scores.** Effect estimates are in colors and FDR value (if ≤0.1) are labeled as numbers. Both protein biomarker predictors and cognitive outcomes were rank normalized to mean 0 and SD 1. Each color block shows the estimated effect for each pair of associations investigated in the primary analyses using linear mixed effect models adjusting for the covariates from Model 1(age, sex, education level, time distance between exam 7 and the neuropsychological testing, and retest indicator). FDR are shown for associations where FDR ≤ 0.1.

$\beta$ = 0.050. After further adjustment using covariates in model 2, we observed similar effects, but with decreased significance (S3 Fig).

**Results from *APOE* $\varepsilon$4 analyses.** No significant differences in effect sizes or direction between *APOE* $\varepsilon$4 carriers and non-carriers was identified by the interaction analysis. The association of higher MPO level with poorer SIM test scores observed in the full sample was also significant in the non-carriers ($\beta$ = −0.072, *SE* = 0.02, *FDR* = 0.036, see S4 Fig). In the carriers, this association was in the same direction but did not reach significance.

**Sensitivity analysis.** When restricting to the sample with neuropsychological testing within 2 years before or after the Offspring exam 7, we observed the same negative association with less significant FDR and an additional positive association between CXCL16 and Trails making test (see S5 Fig).

## Biomarkers and brain MRI measures

We observed significant associations between CD14, CD5L, and sRAGE and total brain volume adjusting for Model 1 covariates (Fig 3).

Higher levels of CD14 and CD5L were significantly associated with smaller TCBV ($\beta$ = −0.14 and −0.13), respectively; see S4 Table for all effect estimates, SEs, and FDRs). Conversely, higher levels sRAGE were significantly associated with larger TCBV ($\beta$ = 0.19). The associations between CD5L and sRAGE with TCBV remained significant after adjusting for Model 2 covariates (S6 Fig).

**Results from *APOE* $\varepsilon$4 analyses.** No significant *APOE* $\varepsilon$4 carrier status interactions were observed, indicating no significant difference between the carriers versus non-carriers in effect

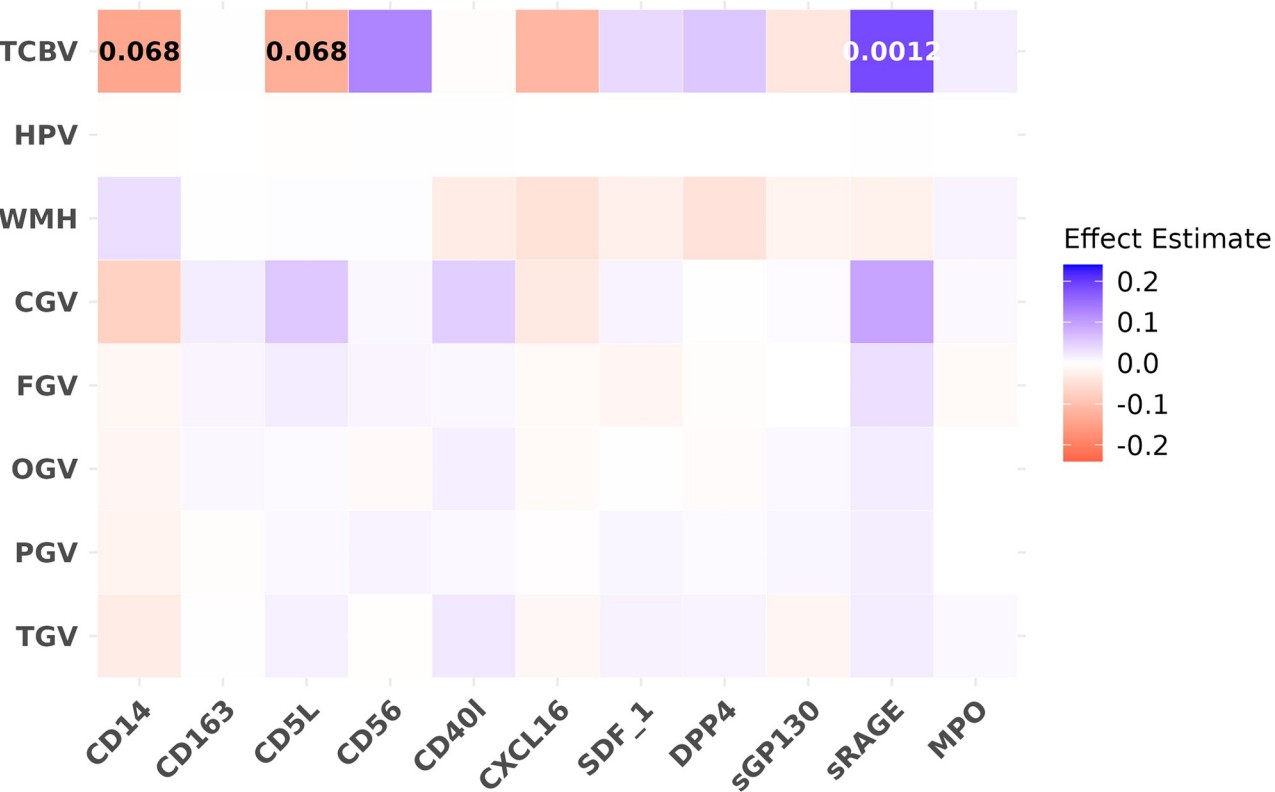

**Fig 3. Effect estimates and FDR value for the associations of protein biomarkers with brain MRI measures.** Effect estimates are in colors and FDR value (if ≤0.1) are labeled as numbers. Protein biomarker predictors were rank normalized to mean 0 and SD 1. Total and regional brain volumes and WMH volume were as percentage of TCV, WMH was also log transformed. Each color block shows the estimated effect for each pair of associations investigated in the primary analyses using linear mixed effect models adjusting for the covariates from Model 1 (age, age², sex, age-sex interaction, and time distance between exam 7 and the MRI scan).

sizes or directions for associations of 11 protein biomarkers with brain MRI measures. The association of higher sRAGE level with larger TCBV observed in the full sample was more significant in the non-carriers ($\beta = 0.22$, $SE = 0.05$, $FDR = 0.00059$, see S7 Fig); the association was in the same direction but did not reach significance in the carriers.

**Sensitivity analysis.** When restricting to sample with brain MRI measures within 2 years before or after the Offspring exam 7, we obtained the same effect directions and similar effect sizes as the main result; in addition, higher levels CD56 was significantly associated with larger TCBV (see S8 Fig).

*Biomarkers and incident dementia.* None of the 11 protein biomarkers were significantly associated with incident dementia over 10 years of follow up in model 1 (FDR ≤ 0.1, Table 2). Marginal associations ($p \leq 0.05$) were observed for CD5L and CD14 with incident dementia ($HR = 1.20$ per SD unit increase in CD5L level; 95% $CI = (1.03, 1.41)$, $p = 0.02$; ($HR = 1.20$ per SD unit increase in CD14 level; 95% $CI = (1.00, 1.44)$, $p = 0.05$). After additional adjustment for CVD risk factors (Model 2), the effects are reduced and do not reach nominal significance (S5 Table).

**Results from *APOE ε*4 analyses.** There were no statistically significant differences in association between *APOE ε*4 carriers and non-carriers (Fig 4).

**Sensitivity analyses and dementia subtypes.** The hazard ratios for incident AD were similar to the hazard ratios for all cause dementia (Table 3). In addition to the marginal

**Table 2. Association of protein biomarkers with incident all-cause dementia within 10 years of follow-up [a].**

| Biomarker | Hazard Ratio | 95% CI | *p*-value | FDR [b] |
|-----------|--------------|--------|-----------|---------|
| **CD14** | 1.20 | (1.00, 1.44) | **0.05** | 0.25 |
| **CD163** | 1.11 | (0.92, 1.33) | 0.28 | 0.44 |
| **CD5L** | 1.20 | (1.03, 1.41) | **0.02** | 0.25 |
| **CD56** | 1.03 | (0.85, 1.25) | 0.79 | 0.97 |
| **CD40L** | 1.19 | (0.98, 1.45) | 0.08 | 0.25 |
| **CXCL16** | 1.00 | (0.83, 1.20) | 0.97 | 0.99 |
| **SDF1** | 1.17 | (0.98, 1.40) | 0.09 | 0.25 |
| **DPP4** | 0.94 | (0.80, 1.11) | 0.49 | 0.67 |
| **sGP130** | 1.11 | (0.93, 1.31) | 0.24 | 0.44 |
| **sRAGE** | 0.86 | (0.69, 1.08) | 0.19 | 0.42 |
| **MPO** | 1.00 | (0.85, 1.18) | 0.99 | 0.99 |

[a]. Cox proportional hazard regression model adjusting for Model 1 covariates (age, sex, education level).

[b]. *FDR* ≤ 0.1 threshold to account for multiple testing.

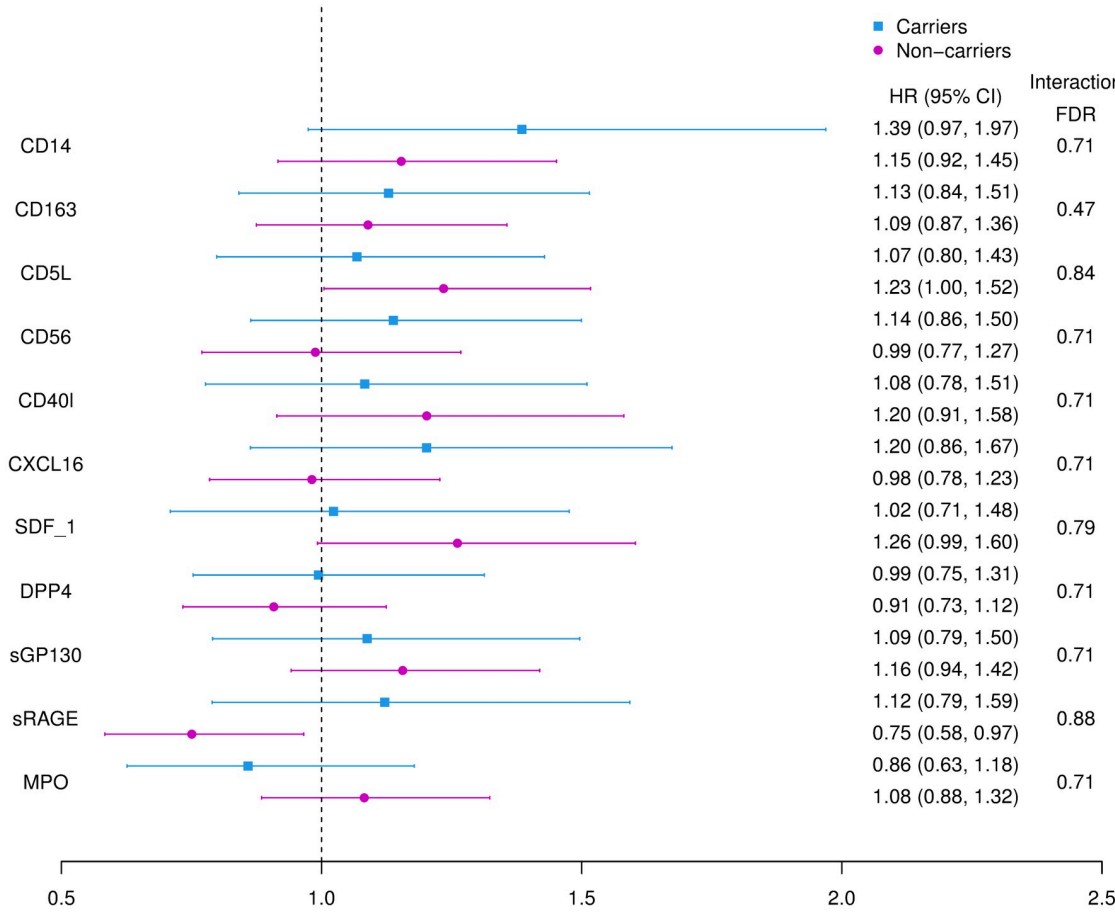

**Fig 4. HR of incident dementia per SD unit higher protein biomarkers, stratified by *APOE ε*4 carrier.** Each color highlights the results from analyses using Cox proportional hazard regression models adjusting for Model 1 covariates (age, sex, and education level) in the carriers versus non-carriers subgroup. Points are effect estimates while lines indicate the 95% *CI. APOE ε*4 carrier status interaction FDR are also listed.

**Table 3. Association of protein biomarkers with incident AD and MCI or dementia within 10-year follow-up [a].**

| Biomarker | Incident AD | | | | Incident MCI or dementia | | | |
|---|---|---|---|---|---|---|---|---|
| | Hazard Ratio | 95% CI | *p*-value | FDR [b] | Hazard Ratio | 95% CI | *p*-value | FDR |
| CD14 | 1.27 | (1.03, 1.57) | **0.02** | 0.15 | 1.06 | (0.92, 1.21) | 0.42 | 0.82 |
| CD163 | 1.09 | (0.88, 1.36) | 0.42 | 0.58 | 1.05 | (0.92, 1.21) | 0.46 | 0.82 |
| CD5L | 1.22 | (1.02, 1.45) | **0.03** | 0.15 | 1.03 | (0.90, 1.18) | 0.68 | 0.82 |
| CD56 | 1.10 | (0.88, 1.36) | 0.40 | 0.58 | 1.02 | (0.88, 1.19) | 0.74 | 0.82 |
| CD40L | 1.24 | (0.98, 1.56) | 0.07 | 0.25 | 1.07 | (0.92, 1.25) | 0.35 | 0.82 |
| CXCL16 | 0.91 | (0.75, 1.10) | 0.35 | 0.58 | 0.98 | (0.86, 1.11) | 0.74 | 0.82 |
| SDF1 | 1.17 | (0.93, 1.46) | 0.17 | 0.47 | 1.12 | (0.98, 1.28) | 0.09 | 0.48 |
| DPP4 | 0.96 | (0.78, 1.18) | 0.69 | 0.69 | 0.96 | (0.85, 1.10) | 0.59 | 0.82 |
| sGP130 | 1.06 | (0.87, 1.29) | 0.58 | 0.69 | 0.95 | (0.84, 1.08) | 0.44 | 0.82 |
| sRAGE | 0.87 | (0.69, 1.11) | 0.27 | 0.58 | 0.88 | (0.77, 1.01) | 0.06 | 0.48 |
| MPO | 0.96 | (0.79, 1.17) | 0.67 | 0.69 | 1.00 | (0.89, 1.13) | 0.94 | 0.94 |

[a]. Cox proportional hazard regression model adjusting for Model 1 covariates (age, sex, education level).

[b]. FDR ≤ 0.1 threshold to account for the correlation between multiple testing.

association between increased CD14 with higher risk of AD, increased CD5L was also marginally associated with higher risk of incident AD within 10 years of follow-up (*HR* = 1.22, 95% *CI* = (0.88, 1.36), *p* = 0.03, *FDR* = 0.15). The hazard ratios for all cause dementia plus MCI tended to be closer to 1.0 than for all cause dementia alone.

## Discussion

Blood-based biomarkers for cognitive decline and dementia are of high interest due to their low cost and lack of need for invasive procedures. Therefore, this field of study is highly active [64, 65]. We observed several findings from our investigation of the association between 11 circulating inflammatory biomarkers with neuropsychological test performance, structural MRI-derived volumetric indices, and risk of incident dementia in a large community-based cohort of older adults. First, higher levels of CD40L and MPO were associated with poorer performance on neuropsychological tests of attention and executive function. Our findings were robust to adjustment of *APOE* ε4 carrier status and CVD risk factors and the effects were not significantly different in *APOE* ε4 carriers compared to non-carriers. Second, higher CD5L, and CXCL16 levels were associated with smaller total brain volumes, whereas higher CD56 and sRAGE were associated with larger total brain volume. The associations of CD5L and sRAGE with total brain volume persisted after adjustment for *APOE* ε4 carrier status and CVD risk factors and there was no difference in associations in *APOE* ε4 carriers vs non-carriers. Finally, we observed nominally significant associations between higher CD5L and increased dementia risk, but these did not achieve FDR≤0.1. As previously reported in FHS [4] higher levels of CD14 were associated with lower scores on the Similarities test. While we observed significant associations with cognitive scores and MRI brain volumes, we found no significant associations between these biomarkers and dementia or Alzheimer disease. This may be due to the limited number of incident cases of AD and dementia in our sample. The cognitive scores and MRI measures are considered endophenotypes for cognitive decline, mild cognitive impairment, and dementia. In a community-based sample such as FHS, these quantitative phenotypes are typically more powerful than the incident outcomes [66].

If confirmed, our findings suggest potential new immunologic pathways and additional peripheral biomarkers of inflammation that may be important for prevention and treatment of dementia and associated premature brain aging.

CD5L is involved in macrophage biology [67] and regulation of T cells, specifically T helper 17 (Th17) cells [68], innate and adaptive immune cells known to be involved in infection, atherosclerosis and cancer. CD5L appears to provide a molecular switch that influences the functional state of the Th17 cells (pathogenic vs. nonpathogenic). Th17 cells play a role in CNS inflammation [69] and vascular and neuronal deficits in mouse models of post-infectious encephalitis [70]. A small case control study in humans demonstrates an association between Th17 cells and AD dementia [71]. Additionally, CD5L is expressed by infiltrating macrophages in the brain after stroke, where it is thought to bind to damage-associated molecular patterns (DAMP), facilitating the phagocytosis of DAMPs and DAMP-associated dead cells, thereby preventing the binding to toll-like receptors and membrane bound RAGE (mRAGE). This inhibits the mRAGE/AGE proinflammatory pathway in the brain [14]. Interestingly, CD5L is expressed in retinal microglial cells and may play a role in macular degeneration [72]. In the present study, higher CD5L levels are associated with smaller total brain volume, an MRI marker of brain aging. In addition, higher CD5L was associated with all cause dementia and AD dementia although associations did not meet significance after multiple test correction. If our findings are confirmed in additional studies, measures to reverse increases in CD5L and its impact on related innate and adaptive immune cells may have preventive implications and offer targets for novel treatments for dementia.

In a small case control study in a Finnish population, a MPO polymorphism, associated with increased expression of MPO, in the presence of *APOE ɛ*4, increased risk for AD and decreased age at onset of AD in men [73]. Mouse models of AD with MPO deficiency (5XFAD-MPO KO) demonstrate improved cognitive functioning and hippocampal staining demonstrates lower inflammatory mediators, mRNA levels showed reduced *APOE* but there were no difference in amyloid-beta plaques [74]. Together the mouse model results suggest a role for MPO in the pathology of AD. In a small case-control study of recurrent depressive disorder, MPO expression associates with measures of cognitive function including executive function (Trail making test, Stroop test), verbal fluency, and auditory-verbal learning [75]. Our study results extend this evidence to a large general population sample observing an association between MPO and a neuropsychological test of attention and executive function (Similarities) that persisted after adjustment for important confounders and *APOE* status. We did not identify an interaction with *APOE ɛ*4 but our analyses were underpowered to detect modest differences in effect between carriers and non-carriers.

CD40 and its ligand, CD40L regulate T and B lymphocytes and innate immune cell activation. In the central nervous system, CD40 expression is found on many cells including microglia [76]. CD40L is implicated in the neuroinflammation of AD dementia pathogenesis by working together with amyloid-beta peptides to promote pro-inflammatory responses in the brain leading to neuronal death [77]. In this study, we observed that increased levels of CD40L measured in the peripheral circulation were associated with poorer performance on a neuropsychological test of attention and executive function.

Advanced glycation end products (AGEs) have been linked to AD dementia pathology including amyloid, tau and neurofibrillary tangles in the brain [9]. *APOE ɛ*4 carriers may have higher levels of glycation compared to non-carriers increasing their risk for dementia [78]. sRAGE may play a protective anti-inflammatory role, acting as a decoy receptor for the binding of AGEs to membrane bound RAGE (mRAGE), thus decreasing the proinflammatory response of the AGE/mRAGE system [9]. Higher sRAGE was associated with prevalent but not incident dementia in a large Dutch population raising the question of a short-term

association and the need for more prospective studies [17]. Our analyses, while not statistically significant, suggest marginal associations for higher level of sRAGE with lower risk of MCI or dementia. Our cross-sectional analyses provide further support as higher sRAGE was significantly associated with larger total and regional brain volumes.

Strengths of this study include the use of data from the FHS, a community-based sample in which cognitive functioning was well characterized with a wide array of neuropsychological tests relative to other cohorts and quantitative MRI techniques were used. Also, dementia cases were reviewed and adjudicated using a standard protocol by neurologist and neuropsychologist. Our study also leverages the benefit of the longitudinal cohort design of FHS, composed of participants free of dementia at the time that the protein biomarkers were assayed and followed for clinical diagnosis of dementia, AD dementia and MCI, so that risk of these neurodegenerative diseases can be studied. All covariates and potential confounders specified in our analysis, such as presence of diabetes, and prevalent CVD and AF, were directly measured or validated with medical records in the FHS. In addition, we were able to include *APOE* genotype and used analytical methods to account for familial correlation between participants from the FHS. Limitations of our study include, first, the neuropsychological and brain MRI analyses are cross-sectional, so we cannot infer causality or directionality. In addition, neuropsychological testing and brain MRI did not occur at the same time as the blood draw used for measuring the protein biomarkers, which may result in higher variability of our estimates. Therefore, we included a covariate of the time between blood draw and assessment of neurological outcomes in our models and performed a sensitivity analysis including only exams within a closer, 2-year time period. Second, our study is limited by the sample size; the limited number of cases of incident dementia, AD dementia, and MCI lead to low power to detect moderate associations with these outcomes. The inflammatory proteins were measured in plasma, the circulating levels of these markers may not reflect what is found in the cerebrospinal fluid and brain but offer potential markers that can be easily obtained. Finally, the FHS is composed of individuals that primarily are white; therefore, the results may not be generalizable to persons of other races or ethnic backgrounds.

## Conclusion

This study, conducted in a large well-characterized community-based sample, suggests associations between protein biomarkers of inflammation related to innate and adaptive immune cells with both cognitive and brain MRI measures of brain aging and incident dementia. In addition to CD14, CD40L, and MPO relate to the executive function neuropsychological testing domain; CD5L and sRAGE relate to global and regional MRI markers of brain atrophy; CD5L shows marginal association with dementia, and AD dementia. Our study suggests that these peripheral inflammatory factors, which are generally produced in response to innate immune activation, may play a role in the pathophysiology of cognitive decline and AD. While further confirmation of our findings in a larger diverse sample is needed, this data indicates potential immune pathways for future exploration into early risk factors for dementia and AD (ie bacterial/viral exposures) and potential treatments.

## Supporting information

**S1 Table.** a). Lower and upper detectable limits and the upper CV limit of the protein biomarkers included. Values outside the detectable limits were set to the lower or upper detectable limits respectively and were set to unknown if the CV was above the threshold. [a] a. Details on protein biomarker measurement and quality control have been described in Ho JE, Lyass A, Courchesne P, Chen G, Liu C, Yin X, et al. Protein Biomarkers of Cardiovascular Disease and

Mortality in the Community. *Journal of the American Heart Association*: *Cardiovascular and Cerebrovascular Disease*. 2018 Jul 1;7(14). doi:10.1161/JAHA.117.008108. b). Number of cases where the protein biomarkers are below, within, or above the detectable limits, or are set to missing because CV was above the upper limit threshold or are missing. [a] a. Sample sizes for all biomarkers are different due to the different number of missingness. [b]. Not all individuals were included for all protein assays.
(PDF)

**S2 Table.** a). Distribution of participants protein biomarkers in the dementia, neuropsychological test, and MRI study samples. b). Participants' neuropsychological testing scores and brain MRI measures stratified by APOE ε4 carrier status.
(PDF)

**S3 Table. Cross-sectional association of protein biomarkers with neuropsychological test performance using linear mixed effect models adjusting for Model 1 covariates [a].** Both protein biomarkers and neuropsychological test scores are rank normalized to mean 0 and standard deviation 1. a. Model 1 covariates (age, sex, education level, time distances between exam 7 and the Neuropsychological testing, and retest indicator) were included. b. $FDR \leq 0.1$ threshold to account for multiple testing.
(PDF)

**S4 Table. Cross-sectional association of protein biomarkers with brain MRI measures using linear mixed effect models adjusting for Model 1 covariates [a].** Protein biomarkers are rank normalized to mean 0 and standard deviation 1. a. Model 1 covariates (age, age2, sex, age-sex interaction, time distance between exam 7 and the brain MRI scan) were included. Total and regional brain volumes and WMH volume were as percentage of total cranial volume, WMH was also log transformed. b. $FDR \leq 0.1$ threshold to account for multiple testing.
(PDF)

**S5 Table. Sensitivity analyses results: Association between protein biomarkers and incident all-cause dementia within 10 years follow-up using cox proportional hazard regression models adjusting for Model 2 covariates [a].** a. Model 2 covariates includes Model 1 covariates with additional adjustment for *APOE* ε4 carrier status and CVD risk factors (SBP, treatment for hypertension, BMI, current smoking status, total cholesterol levels, HDL, presence of diabetes, prevalent AF, and prevalent CVD). b. FDR $\leq 0.1$ threshold to account for multiple testing.
(PDF)

**S1 Fig. Overlapping in the neuropsychological test, MRI, and dementia study samples.** Numbers labeled are number of participants included in each scenario. There are 1038 participants included in all three subsamples; 2094 in both neuropsychological test and MRI subsample; 1171 in both neuropsychological test and dementia subsample; 1040 in both MRI and dementia subsample. Participants included in the neuropsychological test and MRI subsamples but excluded from the dementia sample were due to age being greater than to equal to 60 years at exam 7. Participants included in the dementia subsample but not the neuropsychological test or MRI subsamples were due to missing records of neuropsychological testing, or MRI, or both measures within 5 years of exam 7.
(TIF)

**S2 Fig. Correlations among the cognitive scores and MRI measurements in the overlap between the cognitive test sample and the MRI sample (n = 2094).** Participants may take cognitive tests and brain MRI measures on different dates. For the 2094 participants included

in both the cognitive test and brain MRI samples, there are 2002 participants have these two measures on the same day; among the other 92 participants, 81 have cognitive test prior to the brain MRI measures. The mean difference in dates between the cognitive test and brain MRI measures for those 92 participants is 1.03 years with a standard deviation of 1.23 years.
(TIF)

**S3 Fig. Effect estimates and significant FDR for the cross-sectional associations of protein biomarkers with neuropsychological testing scores, Model 2 covariates.** Effect estimates are in colors and FDR value (if ≤0.1) are labeled as numbers. Both protein biomarker predictors and cognitive outcomes were rank normalized to mean 0 and SD 1. Each color block shows the estimated effect for each pair of associations investigated in the primary analyses using linear mixed effect models adjusting for the covariates from Model 2 (Model 1 covariates plus APOE ε4 carrier status and CVD risk factors: SBP, treatment for hypertension, BMI, current smoking status, total cholesterol levels, HDL, presence of diabetes, prevalent AF, and prevalent CVD). FDR are shown for associations where FDR ≤ 0.1.
(TIF)

**S4 Fig. Effect estimates and significant FDR for the cross-sectional associations of protein biomarkers with neuropsychological testing scores stratified by *APOE ε4* carrier status.** Effect estimates are in colors and FDR value (if ≤0.1) are labeled as numbers. Both protein biomarker predictors and cognitive outcomes were rank normalized to mean 0 and SD 1. Each color block shows the estimated effect for each pair of associations investigated in the primary analyses using linear mixed effect models adjusting for the covariates from Model 1. FDR are shown for associations where FDR ≤ 0.1.
(TIF)

**S5 Fig. Effect estimates and significant FDR for the cross-sectional associations of protein biomarkers with neuropsychological testing scores on the subsample containing participants who had neuropsychological testing within 2 years before or after exam 7 (n = 2185).** Effect estimates are in colors and FDR value (if ≤0.1) are labeled as numbers. Both protein biomarker predictors and cognitive outcomes were rank normalized to mean 0 and SD 1. Each color block shows the estimated effect for each pair of associations investigated in the primary analyses using linear mixed effect models adjusting for the covariates from Model 2. FDR are shown for associations where FDR ≤ 0.1.
(TIF)

**S6 Fig. Effect estimates and significant FDR for the cross-sectional associations of protein biomarkers with brain MRI measures, Model 2 covariates.** Effect estimates are in colors and FDR value (if ≤0.1) are labeled as numbers. Protein biomarker predictors were rank normalized to mean 0 and SD 1. Total and regional brain volumes and WMH volume were as percentage of total cranial volume, WMH was also log transformed. Each color block shows the estimated effect for each pair of associations investigated in the primary analyses using linear mixed effect models adjusting for the covariates from Model 2 (Model 1 covariates plus APOE ε4 carrier status and CVD risk factors: SBP, treatment for hypertension, BMI, current smoking status, total cholesterol levels, HDL, presence of diabetes, prevalent AF, and prevalent CVD). FDR are shown for associations where FDR ≤ 0.1.
(TIF)

**S7 Fig. Effect estimates and significant FDR for the cross-sectional associations of protein biomarkers with brain MRI measures stratified by the *APOE ε4* carrier status.** Effect estimates are in colors and FDR value (if ≤0.1) are labeled as numbers. Protein biomarker

predictors were rank normalized to mean 0 and SD 1. Total and regional brain volumes and WMH volume were as percentage of total cranial volume, WMH was also log transformed. Each color block shows the estimated effect for each pair of associations investigated in the primary analyses using linear mixed effect models adjusting for the covariates from Model 1. FDR are shown for associations where FDR ≤ 0.1.
(TIF)

**S8 Fig. Effect estimates and significant FDR for the cross-sectional associations of protein biomarkers with brain MRI measures on the subsample containing participants who had brain MRI measures within 2 years before or after exam 7 (n = 1945).** Effect estimates are in colors and FDR value (if ≤0.1) are labeled as numbers. Protein biomarker predictors were rank normalized to mean 0 and SD 1. Total and regional brain volumes and WMH volume were as percentage of total cranial volume, WMH was also log transformed. Each color block shows the estimated effect for each pair of associations investigated in the primary analyses using linear mixed effect models adjusting for the covariates from Model 1. FDR are shown for associations where FDR ≤ 0.1.
(TIF)

## Acknowledgments

The authors thank the Framingham Heart Study participants, as well as the study team for their contributions.

## Author Contributions

**Conceptualization:** Margaret F. Doyle, Joanne M. Murabito, Kathryn L. Lunetta.

**Data curation:** Yuan Fang.

**Formal analysis:** Yuan Fang.

**Funding acquisition:** Margaret F. Doyle, Michael L. Alosco, Joanne M. Murabito, Kathryn L. Lunetta.

**Investigation:** Yuan Fang, Margaret F. Doyle, Jiachen Chen, Michael L. Alosco, Jesse Mez, Claudia L. Satizabal, Wei Qiao Qiu, Joanne M. Murabito, Kathryn L. Lunetta.

**Validation:** Yuan Fang, Margaret F. Doyle, Jiachen Chen, Michael L. Alosco, Jesse Mez, Claudia L. Satizabal, Wei Qiao Qiu, Joanne M. Murabito, Kathryn L. Lunetta.

**Visualization:** Yuan Fang.

**Writing – original draft:** Yuan Fang, Margaret F. Doyle, Joanne M. Murabito, Kathryn L. Lunetta.

**Writing – review & editing:** Yuan Fang, Margaret F. Doyle, Jiachen Chen, Michael L. Alosco, Jesse Mez, Claudia L. Satizabal, Wei Qiao Qiu, Joanne M. Murabito, Kathryn L. Lunetta.

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
