## [Decision Letter · Decision Letter 0]

18 Jul 2022

PONE-D-22-16494Association between inflammatory biomarkers and cognitive agingPLOS ONE

Dear Dr. Fang,

Thank you for submitting your manuscript to PLOS ONE. After careful consideration, we feel that it has merit but does not fully meet PLOS ONE’s publication criteria as it currently stands. Therefore, we invite you to submit a revised version of the manuscript that addresses the points raised during the review process.

We look forward to receiving your revised manuscript.

Kind regards,

Muhammad Tarek Abdel Ghafar, M.D

Academic Editor

PLOS ONE

Journal Requirements:

Reviewers' comments:

Reviewer's Responses to Questions

**Comments to the Author**

1. Is the manuscript technically sound, and do the data support the conclusions?

Reviewer #1: Yes

Reviewer #2: Yes

Reviewer #3: Partly

2. Has the statistical analysis been performed appropriately and rigorously? 

Reviewer #1: Yes

Reviewer #2: Yes

Reviewer #3: Yes

3. Have the authors made all data underlying the findings in their manuscript fully available?

Reviewer #1: Yes

Reviewer #2: Yes

Reviewer #3: Yes

4. Is the manuscript presented in an intelligible fashion and written in standard English?

Reviewer #1: Yes

Reviewer #2: Yes

Reviewer #3: No

5. Review Comments to the Author

Reviewer #1: Dear authors, I admire the efforts done in the manuscript for improving the future understanding of the pathophysiology of dementia. I would like to recommend adding references relating to the vascular regional hypo-perfusion as an associated substrate for accelerating the cognitive decline, whether in this manuscript or future ones. One example of a recent reference can be found here:

Wu YT, Bennett HC, Chon U, Vanselow DJ, Zhang Q, Muñoz-Castañeda R, Cheng KC, Osten P, Drew PJ, Kim Y. Quantitative relationship between cerebrovascular network and neuronal cell types in mice. Cell Rep. 2022 Jun 21;39(12):110978. doi: 10.1016/j.celrep.2022.110978. PMID: 35732133.

Reviewer #2: The authors examine the association of 11 plasma proteins (CD14, CD163, CD5L, CD56, CD40L, CXCL16, SDF1, DPP4, SGP130, sRAGE, and MPO) related to immune and inflammatory responses with measures of cognitive function, brain MRI and dementia risk. And results are showing significant associations between higher levels of CD40L and myeloperoxidase (MPO) with executive dysfunction; CD5L levels and sRAGE with brain volumes, APOE ε4 carrier status with cardiovascular risk factors that indicates the involvement of an activated immune system in abnormal brain aging.

Excellent research on an extraordinarily important subject and very striking results are clearly explained.

Reviewer #3: Research Summary

This is a biomarker development study where the author(s) have attempted to investigate the association of inflammatory biomarkers with neuropsychological test results and brain MRI result outcomes in FHS offspring cohort study. 11 protein biomarkers were chosen for the study and their levels were estimated in the frozen fasting plasma samples using sandwich ELISA methods. Their finding suggested co-relation of several inflammatory markers such as CD14, CD5L and MPO with exclusive cognitive dysfunction. Additionally, CD5L and sRAGE has also shown co-relations with TCBV. However, none of these biomarkers were significantly associated with increased risk of dementia. Moreover, their study highlights a correlation of the immune system activation with aging associated changes in the brain and cognitive decline.

Comments:

This paper describes an important aspect of involvement of peripheral immune system in ageing. This approach could lead to development of new inflammatory biomarkers associated with ageing and dementia/AD. However, the paper has been written poorly, and it will require a significant amount of work. Importantly, the poor writing makes it highly complicated to read. The paper also failed to describe the experimental model and design of the study. The flow of the paper is also poorly organized. Moreover, the study design, methods and supporting data are disorganized and lack clarity.

Major Comments:

1. The paper requires a major reorganization in terms of writing introduction, elaborated method, discussion and figure generation (Both main figure and supplementary figure; It requires consistency with font size, and organization). It would be helpful for the readers to find required information on the same figure that is explaining similar results.

2. The studied groups is highly diversified which includes APOE carrier and non-carrier; Chronic Leukemia or Lymphoma, patients taking different immunosuppressant. The author has to justify what were the inclusion criteria for such a diverse group. Additionally, for assessing the inflammatory markers why lymphoma, leukemia, or cases with immunosuppressant supplement were considered in study. (Note- Immune systems are either hyperactive or compromised in such situation).

3. It would be necessary to draw a co-relation between cognitive tests and MRI brain volume between entire cohort and the AD/dementia cases.

4. The introduction section and discussion section has been strongly written in favor of association of inflammatory markers and risks of dementia while the results does not reveals a major correlations. Author should elaborate the justifications on this point in the discussion.

5. Authors should include several points in the discussion; 1) which other biomarkers are previously studied or established in ageing, dementia/AD. 2.) What information is available/established in terms of biomarkers in CSF in ageing, dementia/AD?

Minor Comments:

1. Author(s) should cross-check references and possibly add a small description of method even though the citations are given.

Conclusion:

The paper requires a major revision.

6. PLOS authors have the option to publish the peer review history of their article (what does this mean?). If published, this will include your full peer review and any attached files.

Reviewer #1: **Yes: **Mohamed Mostafa

Reviewer #2: **Yes: **Bilgehan A. Acar

Reviewer #3: No

---

## [Author Response · Author response to Decision Letter 0]

5 Aug 2022

Response to Reviewer Comments

We thank the editor and three reviewers for their valuable comments and constructive feedback. Following their suggestions, we have made significant changes to the latest version of the manuscript, as detailed below. The reviewers' comments have been listed, followed by our responses. Page numbers mentioned in the manuscript changes are corresponding to the unmarked version of our revised manuscript without tracked changes.

Reviewer #1: Dear authors, I admire the efforts done in the manuscript for improving the future understanding of the pathophysiology of dementia. I would like to recommend adding references relating to the vascular regional hypo-perfusion as an associated substrate for accelerating the cognitive decline, whether in this manuscript or future ones. One example of a recent reference can be found here:

Wu YT, Bennett HC, Chon U, Vanselow DJ, Zhang Q, Muñoz-Castañeda R, Cheng KC, Osten P, Drew PJ, Kim Y. Quantitative relationship between cerebrovascular network and neuronal cell types in mice. Cell Rep. 2022 Jun 21;39(12):110978. doi: 10.1016/j.celrep.2022.110978. PMID: 35732133.

Response: We thank the reviewer for the positive remarks and for the recommendation of great work as a reference. We recognize that Alzheimer’s disease and cognitive decline are associated with global and regional cerebral hypoperfusion and will certainly consider the relation of inflammatory biomarkers with this in future work.

Reviewer #2: The authors examine the association of 11 plasma proteins (CD14, CD163, CD5L, CD56, CD40L, CXCL16, SDF1, DPP4, SGP130, sRAGE, and MPO) related to immune and inflammatory responses with measures of cognitive function, brain MRI and dementia risk. And results are showing significant associations between higher levels of CD40L and myeloperoxidase (MPO) with executive dysfunction; CD5L levels and sRAGE with brain volumes, APOE ε4 carrier status with cardiovascular risk factors that indicates the involvement of an activated immune system in abnormal brain aging.

Excellent research on an extraordinarily important subject and very striking results are clearly explained.

Response: We thank the reviewer for their time reading the manuscript and for the positive remarks.

Reviewer #3: Research Summary

This is a biomarker development study where the author(s) have attempted to investigate the association of inflammatory biomarkers with neuropsychological test results and brain MRI result outcomes in FHS offspring cohort study. 11 protein biomarkers were chosen for the study and their levels were estimated in the frozen fasting plasma samples using sandwich ELISA methods. Their finding suggested co-relation of several inflammatory markers such as CD14, CD5L and MPO with exclusive cognitive dysfunction. Additionally, CD5L and sRAGE has also shown co-relations with TCBV. However, none of these biomarkers were significantly associated with increased risk of dementia. Moreover, their study highlights a correlation of the immune system activation with aging associated changes in the brain and cognitive decline.

Comments:

This paper describes an important aspect of involvement of peripheral immune system in ageing. This approach could lead to development of new inflammatory biomarkers associated with ageing and dementia/AD. However, the paper has been written poorly, and it will require a significant amount of work. Importantly, the poor writing makes it highly complicated to read. The paper also failed to describe the experimental model and design of the study. The flow of the paper is also poorly organized. Moreover, the study design, methods and supporting data are disorganized and lack clarity.

Response: We thank the reviewer for reading the manuscript carefully and for providing specific comments to improve it. We have made changes as listed below addressing the reviewer’s comments.

Major Comments:

1. The paper requires a major reorganization in terms of writing introduction, elaborated method, discussion and figure generation (Both main figure and supplementary figure; It requires consistency with font size, and organization). It would be helpful for the readers to find required information on the same figure that is explaining similar results.

Response: We have reorganized and reworded the methods and results sections to polish our presentation and edited the discussion. Rewording of the methods and results sections are listed below. Please refer to our response to comments 4 and 5 for specific changes in the discussion sections. We also re-created the figures to make sure that the color scales are the same for both within and across the outcome sets, i.e., the same color refers to the same effect sizes. The colors of the text in the figures are adjusted to make sure we have large contrast. We have made sure that the text within each figure is consistent in font size. We note that the figures uploaded to the PLOS ONE submission system are required to be in high-resolution vector figures format so that all figures can be adjusted to any size without losing resolution. We have limited the information in each of our figures to reflect separate sets of analyses that deliver distinct information.

Manuscript changes: Method section, statistical analyses page 12, page 14; Results section page 15-16, figures including figure names and legends for Fig 2 on page 18, Fig 3 on page 19, and Fig 4 on page 21. Supplementary figures names and legends on pages 45 – 47. 

2. The studied groups is highly diversified which includes APOE carrier and non-carrier; Chronic Leukemia or Lymphoma, patients taking different immunosuppressant. The author has to justify what were the inclusion criteria for such a diverse group. Additionally, for assessing the inflammatory markers why lymphoma, leukemia, or cases with immunosuppressant supplement were considered in study. (Note- Immune systems are either hyperactive or compromised in such situation).

Response: We agree with the reviewer that we made our study more complex than it needed to be. In response, we changed our primary analysis sample to exclude the participants with chronic leukemia or lymphoma and participants who reported taking glucocorticoids at exam 7 (n=~35), and removed the previous sensitivity analyses regarding these participants. With this change, our results are very similar to the previous submission.

Manuscript changes: Methods section on page 8, page 10, and page 11, we updated the inclusion exclusion criteria addressing excluding these participants from the samples; pages 14-15, we updated the sensitivity analyses. Results section pages 16 – 23, we updated the analyses results accordingly including the three tables. Supplementary tables 1 – 5, we deleted the previous supplementary table 6; supplementary figures 1 and 3 – 8, we deleted the previous supplementary figure 9. Supplementary tables and figures captions are updated on the manuscript pages 43 – 47.

3. It would be necessary to draw a co-relation between cognitive tests and MRI brain volume between entire cohort and the AD/dementia cases.

Response: We created correlation plots between cognitive tests and the brain MRI volumes for the study samples we have. We also created correlation plots across the cognitive test and brain MRI measures for the participants who had both cognitive testing and MRI data. The Framingham Heart Study (FHS) participants may take cognitive tests and brain MRI measures on different dates. For the 2094 participants included in both the cognitive test and brain MRI samples, there are 2002 participants that have these two measures on the same day; among the other 92 participants, 81 have cognitive tests prior to the brain MRI measures. The mean difference in dates between the cognitive test and brain MRI measures for those 92 participants is 1.03 years with a standard deviation of 1.23 years. The MRI brain volume measures are highly correlated with each other, but not with white matter hyperintensity. The cognitive scores are moderately correlated with each other. Smaller values in the Trails making tests mean better cognitive test performance whereas in other tests higher scores mean better; hence there are negative correlations between the Trails and others. The MRI measures have low correlations with the cognitive scores, suggesting that the two outcome sets provide independent information. As the test scores and MRIs were performed prior to AD or dementia diagnosis, comparing AD/dementia and controls is not possible. 

We have included the correlation plots in the supplementary materials (and in the attached rebuttal letter for the reviewer), and now mention them in the results.

Manuscript changes: Results section, we mentioned the correlation between the cognitive tests and the brain MRI volumes on page 17. Supplementary figure 2.

4. The introduction section and discussion section has been strongly written in favor of association of inflammatory markers and risks of dementia while the results does not reveals a major correlations. Author should elaborate the justifications on this point in the discussion.

Response: We agree that the lack of associations with dementia in our data compared to the cognitive testing and MRI measures is disappointing. This may be due to the limited number of incident cases of AD and dementia in our sample. The cognitive scores and MRI measures are considered endophenotypes for cognitive decline, mild cognitive impairment, and dementia. In a community-based sample such as FHS, these quantitative phenotypes are typically more powerful than the incident outcomes [1]. 

Manuscript changes: We have added the following to the end of the first paragraph of the discussion on page 24:

“While we observed significant associations with cognitive scores and MRI brain volumes, we found no significant associations between these biomarkers and dementia or Alzheimer’s disease. This may be due to the limited number of incident cases of AD and dementia in our sample. The cognitive scores and MRI measures are considered endophenotypes for cognitive decline, mild cognitive impairment, and dementia. In a community-based sample such as FHS, these quantitative phenotypes are typically more powerful than the incident outcomes [66]" 

5. Authors should include several points in the discussion; 1) which other biomarkers are previously studied or established in ageing, dementia/AD. 

Response: 

We agree that the question of what other blood biomarkers are studied or established in aging and dementia is of interest. However, we do not feel that this could be adequately addressed with a few sentences in the discussion of our paper. We note that several review articles cover aspects of this topic, in particular: 

[2] Olsson B, Lautner R, Andreasson U, Öhrfelt A, Portelius E, Bjerke M, Hölttä M, Rosén C, Olsson C, Strobel G, Wu E, Dakin K, Petzold M, Blennow K, Zetterberg H. CSF and blood biomarkers for the diagnosis of Alzheimer's disease: a systematic review and meta-analysis. Lancet Neurol. 2016 Jun;15(7):673-684. doi: 10.1016/S1474-4422(16)00070-3. Epub 2016 Apr 8. PMID: 27068280.

[3] Altuna-Azkargorta M, Mendioroz-Iriarte M. Blood biomarkers in Alzheimer's disease. Neurologia (Engl Ed). 2021 Nov-Dec;36(9):704-710. doi: 10.1016/j.nrleng.2018.03.006. Epub 2020 Feb 19. PMID: 34752348.

Manuscript changes: We have added a brief note to the beginning of our discussion on page 23:

“Blood-based biomarkers for cognitive decline and dementia are of high interest due to their low cost and lack of need for invasive procedures. Therefore, this field of study is highly active [64,65].”

2.) What information is available/established in terms of biomarkers in CSF in ageing, dementia/AD?

Response: While we agree with the reviewer that many promising CSF biomarkers exist, the goal of the present study is to exam potential blood based biomarkers.

Minor Comments:

1. Author(s) should cross-check references and possibly add a small description of method even though the citations are given.

Response: We have checked and made sure that the references are labeled correctly. 

Manuscript changes: References updated where needed. Most reference number changes in the revised paper are due to additional references added during the revision process.

References

1. Reitz C, Mayeux R. Endophenotypes in normal brain morphology and Alzheimer’s disease: a review. Neuroscience. 2009;164: 174–190. doi:10.1016/J.NEUROSCIENCE.2009.04.006

2. Olsson B, Lautner R, Andreasson U, Öhrfelt A, Portelius E, Bjerke M, et al. CSF and blood biomarkers for the diagnosis of Alzheimer’s disease: a systematic review and meta-analysis. Lancet Neurol. 2016;15: 673–684. doi:10.1016/S1474-4422(16)00070-3

3. Altuna-Azkargorta M, Mendioroz-Iriarte M. Blood biomarkers in Alzheimer’s disease. Neurologia (Barcelona, Spain). 2018;36: 704–710. doi:10.1016/j.nrl.2018.03.006

---

## [Decision Letter · Decision Letter 1]

26 Aug 2022

Association between inflammatory biomarkers and cognitive aging

PONE-D-22-16494R1

Dear Dr. Fang,

We’re pleased to inform you that your manuscript has been judged scientifically suitable for publication and will be formally accepted for publication once it meets all outstanding technical requirements.

Kind regards,

Muhammad Tarek Abdel Ghafar, M.D

Academic Editor

PLOS ONE

Additional Editor Comments (optional):

Reviewers' comments:

Reviewer's Responses to Questions

**Comments to the Author**

1. If the authors have adequately addressed your comments raised in a previous round of review and you feel that this manuscript is now acceptable for publication, you may indicate that here to bypass the “Comments to the Author” section, enter your conflict of interest statement in the “Confidential to Editor” section, and submit your "Accept" recommendation.

Reviewer #1: All comments have been addressed

Reviewer #4: All comments have been addressed

2. Is the manuscript technically sound, and do the data support the conclusions?

Reviewer #1: Yes

Reviewer #4: Yes

3. Has the statistical analysis been performed appropriately and rigorously? 

Reviewer #1: Yes

Reviewer #4: Yes

4. Have the authors made all data underlying the findings in their manuscript fully available?

Reviewer #1: Yes

Reviewer #4: Yes

5. Is the manuscript presented in an intelligible fashion and written in standard English?

Reviewer #1: Yes

Reviewer #4: Yes

6. Review Comments to the Author

Reviewer #1: As per the review comment before, the authors shall publish future manuscripts involving the regional hypoperfusion therapy as a probable cause of dementia.

Reviewer #4: This is a valuable research and will written manuscript and the subject is very important

All the reviewers comments are addressed.

7. PLOS authors have the option to publish the peer review history of their article (what does this mean?). If published, this will include your full peer review and any attached files.

Reviewer #1: **Yes: **Mohamed Mostafa

Reviewer #4: No

---

## [Editor Report · Acceptance letter]

31 Aug 2022

PONE-D-22-16494R1 

Association between inflammatory biomarkers and cognitive aging 

Dear Dr. Fang:

I'm pleased to inform you that your manuscript has been deemed suitable for publication in PLOS ONE. Congratulations! Your manuscript is now with our production department. 

Kind regards, 

on behalf of

Prof Muhammad Tarek Abdel Ghafar 

Academic Editor

PLOS ONE